# Nutritional Assessment of *Lactarius drassinus* and *L. controversus* from the Cold Desert Region of the Northwest Himalayas for Their Potential as Food Supplements

**DOI:** 10.3390/jof9070763

**Published:** 2023-07-20

**Authors:** Hom-Singli Mayirnao, Samta Gupta, Sarda Devi Thokchom, Karuna Sharma, Tahir Mehmood, Surinder Kaur, Yash Pal Sharma, Rupam Kapoor

**Affiliations:** 1Department of Botany, University of Delhi, Delhi 110007, India; homsingli08@gmail.com (H.-S.M.); guptasamta0203@gmail.com (S.G.); srd17thkchm@gmail.com (S.D.T.); sharmak210596@gmail.com (K.S.); 2Department of Botany, University of Jammu, Jammu 180016, India; mehmoodt898@gmail.com (T.M.); yashdbm3@yahoo.co.in (Y.P.S.); 3SGTB Khalsa College, University of Delhi, Delhi 110007, India; surinder@sgtbkhalsa.du.ac.in

**Keywords:** Kargil, cold desert, mushrooms, nutritional and nutraceutical properties, gas chromatography–mass spectrometry, food supplements

## Abstract

Kargil is a cold desert with hostile ecological conditions such as low temperature and precipitation, as well as difficult terrains. However, several wild mushrooms thrive well under such an extreme environment. Despite their abundance, the chemical composition of indigenous mushrooms has not been explored. This study aimed to assess the potential of two wild edible mushrooms from Kargil, *Lactarius drassinus* and *Lactarius controversus*, as food supplements by evaluating their nutritional and nutraceutical properties. Nutritional attributes such as total protein, available carbohydrates, soluble sugars, and vitamins were found to be high in the mushroom species. Furthermore, high mineral accumulation and relatively lower antinutrient concentrations resulted in higher bioavailabilities of Zn, Fe, Ca, and Mg. Gas-chromatography–mass-spectrometry-based metabolite profiling revealed that although the two mushroom species showed similar metabolite compositions, their relative concentrations differed. Sugars were the predominant compounds identified in both the species, with sugar alcohols being the major contributor. The second most abundant class of compound in *L. drassinus* was amino acids, with 5-oxoproline as the major contributor. On the other hand, fatty acids were the second most abundant compounds in *L. controversus*, with high oleic and linoleic acid concentrations. In the ultra-performance-liquid-chromatography-based quantification of phenolic compounds, chlorogenic acid was found to be highest in in terms of its concentration in both the mushrooms studied, followed by quercetin dihydrate and gallic acid in *L. drassinus* and *L. controversus*, respectively. Moreover, high antioxidant activities attributable to their high phenol, flavonoid, and carotenoid concentrations were observed. Overall, the two mushrooms offer well-balanced sources of nutritional and nutraceutical compounds, making them healthy foods.

## 1. Introduction

Kargil is a district in the Ladakh Union territory of India, lying in the northwest Himalayas. The district has a peculiar arctic climatic condition coupled with desert climate; therefore, it is called a cold desert. It is connected to the rest of the country through Zoji La pass, which serves as its lifeline. However, the pass remains closed for an average of 150 days during winter because of heavy snowfall [1,2]. Due to road inaccessibility, Kargil faces acute shortage of essential food commodities, especially during winter. Moreover, a combination of climatic factors such as harsh temperatures, dropping to as low as −40 °C in winter to scorching heat in summer; low precipitation; intense solar irradiance; high-velocity winds; orographic barriers; and edaphic conditions including immature, coarse-textured, high pH, and highly permeable soils, makes the region an inhospitable terrain with sparse diversity and low productivity of vegetation [3]. Since the growing period is short and cultivation during winter is not possible except in greenhouses, vegetables remain a rare item in diet leading to malnourishment [3,4]. Nevertheless, the region constitutes an important home for a diverse group of high-altitude wild macrofungi [5], yet their availability is limited to a few months (June to September), as the sporadic fructification occurs only after slight showers of rain. Due to aforementioned factors, exploitation of the already available food resources, such as wild edible mushrooms, is a favourable option to achieve food security for the natives.

Wild mushrooms have received considerable interests and propelling expansion in many parts of the world [6]. In Asian countries, mushrooms have always been highly prized for their value as food and source of medicines. They find use in indigenous therapies, particularly in traditional Chinese medicine [7]. Only in recent decades have European countries showed interest in studying their impacts on human health [8]. Consequently, an arsenal of studies evaluated the biochemical characteristics of mushrooms and revealed their potential as food supplements [9,10,11]. Although a decent body of literature advocates for the nutritional benefits of various species of mushrooms the world over, the same set of findings cannot be conveniently and rigidly extrapolated to species from different geographical regions, since the biochemical constituents are contingent on the environment in which these mushrooms flourish [12,13,14].

The most consumed mushrooms worldwide include *Agaricus bisporus*, *Pleurotus ostreatus*, *Lentinula edodes*, and *Volvariella volvacea*. They are easily cultivable and are characterised by culinary features and high nutritional value [15,16]. In addition to those that are commonly available in the markets, wild mushrooms are gathered and consumed by indigenous communities from across the globe [17]. The exploitation of mushrooms for culinary purposes as well as pharmaceutical demands have reached new heights, thus presenting them to be a paramount food component [18]. In recent years, there has been growing interest to domesticate wild mushrooms that are considered to have potential in the functional food market [19]. Wild mushrooms have a long association with humankind. Owing to their flavour, texture, and desirable taste, they have an established position in international markets and command higher prices than cultivated mushrooms [19,20]. Consequently, mushrooms represent an attractive frontier in health sector. They have been reported as therapeutic foods for illnesses associated with unhealthy lifestyles, such as hypertension, hypercholesterolemia, atherosclerosis, non-alcoholic fatty liver disease, and even various types of cancer [21]. Contemporary research has validated unique properties of bioactive compounds extracted from numerous species of mushrooms used in treating and reducing the severity of various medical conditions, including COVID-19 [22,23]. Mushrooms contain a wide variety of polysaccharides that are often cited for their immunomodulatory, antioxidant, antidepressant, and anticancer properties, among others [24,25,26]. Clinical trials have substantiated mushrooms such as *P. eryngii* and *A. bisporus* as a favourable choice for “healthy snacking” suitable for individuals with an unhealthy metabolism and personalised nutritional needs [27,28,29]. Thus, wild edible mushrooms have become remarkably important in our diet.

Multitudinous studies have documented indigenous knowledge of wild mushrooms of the northwest Himalayas [30,31,32]. Although some regions of Kargil serve as a rich repository of mushrooms, the practice of adding wild mushrooms in diets is uncommon, with the exception of nomadic tribals and Nepali migrants, due to lack of knowledge on mushrooms’ invigorating benefits. Furthermore, studies on mushrooms of Kargil pertaining to diversity are apparently in their pioneer state, and biochemical characterisation is untouched. In this regard, two wild edible mushrooms, *L. drassinus* Verma et al. (endemic to Kargil) and *L. controversus* (Pers.) Pers., collected from Kargil, were evaluated for their nutritional and nutraceutical properties.

*Lactarius*, a genus within the *Russulaceae* family, exhibits an ectomycorrhizal relationship with species of *Salix*, *Populus*, and *Betula* [33,34,35]. Distribution of *L. controversus* spans across three continents, namely, Europe, North America, and Asia. Notably, the majority of the reports are from Europe (the Netherlands, the United Kingdom, Spain, Belgium, Serbia, Norway, Greece, Hungary, Romania), followed closely by North America (United States of America and Canada) [35,36,37,38,39,40,41]. In Asia, reports of *L. controversus* occurrence has been documented in Kazakhstan, Turkey, and India [33,42,43,44,45]. There are limited studies on *L. controversus*, wherein chemical profile, antioxidant, antibacterial, and cytotoxic activities have been studied [41,42]. On the other hand, *L. drassinus* is a recently identified species reported from Drass, Kargil, by Verma et al. [46]. Hence, no biochemical investigation has been performed per se. Considering the emerging use of wild culinary mushrooms, the present study assessed (i) metabolite profile based on gas chromatography-mass spectrometry (GC-MS); (ii) nutritional value: total protein, carbohydrates (total available carbohydrates, total soluble sugars, reducing sugars, non-reducing sugars, and starch), vitamins (C, B_3_, B_6_), minerals, macroelements (N, P, K, S, Mg, Ca), microelements (Na, Cu, Zn, Fe, Al, Ni), and antinutrients (phytates and tannins); and (iii) nutraceutical properties: total phenol, total flavonoid, ultra-performance liquid-chromatography-based quantification of phenols (gallic acid, chlorogenic acid, vanillin, ferulic acid, and cinnamic acid), flavonoids (quercetin dihydrate), and carotenoids (lycopene and β-carotene), and reducing power as well as antioxidant potential of *L. drassinus* and *L. controversus* from Kargil.

## 2. Material and Methods

### 2.1. Collection and Identification

Two wild edible mushrooms, species 1 and species 2, were collected from Tesbo and Holiyal, respectively, in Kargil in August, 2021. Macromorphological features of the fresh basidiocarps and reaction to various chemicals [47] were examined (Appendix A). The Methuen Handbook of Colour [48] was followed for colour indications.

Total genomic DNA was extracted from dried samples following the CTAB method [49]. Molecular identification was performed by selecting the internal transcribed spacer (ITS) gene region of the rRNA and sequenced by employing the Sanger sequencing method.

### 2.2. GC-MS-Based Metabolite Profiling

Dried fruiting bodies (1 g) were pulverised and extracted in 100 mL methanol for 3 h. Following extraction, the solution was filtered using Whatman No. 1 filter paper and dried in rotovap at 45 °C. Derivatisation was performed following the protocol of Zarate et al. [50]. To 20 mg extract, 1 mL of pre-chilled solvent (isopropanol/acetonitrile/water in a ratio of 3:3:2) was added, followed by the internal standard, ribitol (0.02 mg mL^−1^). The mixture was vortexed for 10 s and centrifuged at 12,000 rpm for 10 min, and the supernatant was lyophilised. A total of 50 µL of methoxyamine hydrochloride (20 mg mL^−1^) prepared in pyridine was added following lyophilisation. The sample was trimethylsilylated at 37 °C for 30 min by adding 100 µL MSTFA.

The constituents of extracts were analysed using GCMS-QP2010 Ultra (Shimadzu Corp., Kyoto, Japan). The column oven temperature was programmed initially at 80 °C (held for 2 min) and rose to 280 °C at the rate of 10 °C min^−1^ (held for 18 min). The carrier gas used was He with a 1.21 mL min^−1^ flow rate. A blank analysis was first conducted using 1 µL of solvent. The reconstituted extract solution (1 µL) was used for GC-MS analysis using direct liquid injection with a split ratio of 1:10. Mass spectra were taken at 70 eV with an EI source having a mass range of 40–650 amu. Obtained spectra of the components were compared with the database spectra of known components stored in the GC-MS library using the National Institute of Standards and Technology (NIST-14) and WILEY-08 search. The relative percent composition of components was calculated. Measurement of peak areas and data processing were carried out by GC-MS postrun solution software (Shimadzu Corp., Kyoto, Japan).

### 2.3. Total Protein

Concentration of total protein was determined by Bradford’s method [51]. Pulverised samples (500 mg) were extracted with 10 mL of pre-chilled extraction buffer (50 mM Tris HCl (pH 7.5), 0.5% polyvinyl pyrrolidone, and 200 mM β-mercaptoethanol). Extraction was carried out at 4 °C for 1 h and centrifuged at 2000× *g* for 20 min at 4 °C, and the resulting supernatant was used for the assay. To 100 µL extract, 5 mL of Bradford’s reagent was added and mixed immediately. Absorbance was recorded at 595 nm after 10 min of dark incubation. Protein concentration in was determined from the standard curve of BSA and expressed as mg g^−1^ dry weight (DW).

### 2.4. Total Available Carbohydrates

Concentration of total available carbohydrates (TAC) was performed following the Anthrone method [52]. Samples (100 mg) were digested in 5 mL of 52% HCl for 24 h in the dark and filtered with Whatman No. 1 filter paper. The volume of filtrate was adjusted to 40 mL. To this, 2 mL of 0.1% anthrone solution prepared in 70% H_2_SO_4_ was added, and the mixture was boiled for 10 min. Absorbance was recorded at 630 nm after cooling. A calibration curve obtained from the known concentration of glucose was used to express TAC as mg g^−1^ DW.

### 2.5. Sugars and Starch

Extraction was performed by taking 3 g each of the samples dissolved in 100 mL of 80% ethanol. The samples were boiled for 10 min and centrifuged at 2500 rpm for 5 min after cooling. The supernatant was used for sugar analyses, and the pellet was stored at −20 °C for starch analysis.

Total soluble sugar (TSS) concentration was estimated by taking 2 mL extract and 1 mL PSA reagent (5 mL H_2_SO_4_ in 5% phenol). The test tubes were mixed thoroughly, and absorbance was measured at 490 nm after 20 min incubation at 37 °C [53]. TSS was quantified using a calibration curve obtained from glucose.

Concentrations of reducing sugars (RS) were measured following the DNS method [54]. A total of 1 mL extract was mixed with 1 mL DNS reagent (1.5 g of dinitrosalicylic acid in 30 mL of 2 M NaOH with the volume adjusted to 150 mL with distilled water). The reaction mixture was boiled for 5 min followed by absorbance at 515 nm. Quantification of non-reducing sugars (NRS) was done by subtracting the values of RS from that of TSS.

For estimation of starch, pellets were resuspended in 5 mL distilled water and 6.5 mL 70% HClO_4_ and filtered with Whatman filter paper No. 1. The filtrates were added to 3 mL of concentrated H_2_SO_4_ and allowed to cool, and absorbance was measured at 315 nm. Concentration was estimated using standard curve of glucose [55]. A conversion factor of 0.9 was used to express the starch content.

### 2.6. Phenols and Flavonoids

The extract was prepared in 100 mL 80% ethanol kept for 24 h at room temperature, maintained in a dark condition. Homogenate was centrifuged at 5000× *g* for 10 min and extracted twice. Supernatants were pooled together and air dried. The extracts were redissolved in ethanol (10 mg mL^−1^) for further use.

The Folin–Ciocalteu colorimetric method was used for the determination of total phenol [56]. To 0.5 mL extract, 2 mL Folin–Ciocalteu reagent and 2.5 mL 10% NaHCO_3_ solution were added, followed by 30 min incubation at 40 °C and absorbance recorded at 765 nm. Total phenol was represented in terms of gallic acid equivalent (mg GAE g^−1^ of DW).

The total flavonoid was determined by the aluminium chloride colorimetric method [57]. In 500 µL extracts, 500 µL of 2% AlCl_3_ and 500 µL 1 M CH_3_CO_2_K were added. Absorbance was recorded at 415 nm following incubation at room temperature for 30 min. The results were represented as quercetin equivalent (mg QE g^−1^ DW).

### 2.7. Quantification of Phenolic Compounds Using UPLC

For extraction of phenolic compounds (phenolic acids and flavonoid), pulverised samples were homogenised using HPLC-grade methanol followed by sonication for 5 min, thereafter being placed in water bath at 64 °C for 4 h. The extraction process was repeated two times. The obtained extracts were then combined and concentrated. Extracts were filtered with membrane filters of pore size 0.2 μm and stored at 4 °C until further use.

The extracts were analysed using an Acquity UPLC™ H-Class System (Waters Corp., Milford, CT, USA) equipped with a Quaternary Solvent Manager FTN coupled to a UV-photodiode array detector. Separation was achieved on a BEH C_18_ column (1.7 µm, 2.1 mm × 50 mm) thermostatted at 50 °C with the auto sampler programmed at 5 °C. Injection volume was 2 µL. Total run time was 8 min, with a sequence of isocratic and linear gradient flow rate of 0.45 mL min^−1^. Following the method described by Galani et al. [58], a solvent system consisting of solvent A (0.1% formic acid in water) and solvent B (95% methanol, 5% water, 0.1% formic acid) was used with the following gradient: starting with isocratic elution with 100% solvent A held for 0.5 min and installing a linear gradient to obtain 100% solvent B at 4.5 min held for 1 min. The column was programmed to starting conditions for 0.2 min and re-equilibrated for 1.8 min. For a standard calibration curve, different concentrations of standards for gallic acid, chlorogenic acid, vanillin, ferulic acid, quercetin dihydrate, and cinnamic acid were used. Compounds were identified based on the retention times of standards, and quantification was achieved by absorbance recorded in the chromatograms relative to external standards at 280 nm with a data acquisition rate of 20 points s^−1^. Waters Empower^®^ 3 software was used for chromatographic data gathering and integration. All standard curves showed high degrees of linearity (r^2^ > 0.999).

### 2.8. Carotenoids

For estimation of carotenoids, β-carotene and lycopene, samples were cold extracted with 10 mL acetone–hexane mixture (4:6) and filtered through Whatman No. 4 after vigorous shaking for 10 min. Optical density was measured at 453, 505, 645, and 663 nm [59]. β-Carotene and lycopene concentrations were calculated using the following equations:β-carotene (µg g^−1^ DW) = 0.216 A_663_ − 1.22 A_645_ − 0.304 A_505_ + 0.452 A_453_
Lycopene (µg g^−1^ DW) = −0.0458 A_663_ + 0.204 A_645_ + 0.372 A_505_ − 0.0806 A_453_

### 2.9. Antioxidant Potential

#### 2.9.1. Reducing Power

Reducing power was estimated by taking 2.5 mL of sample (0.05–0.2 mg mL^−1^), 2.5 mL of 0.1 M sodium phosphate buffer (pH 6.6), and 2.5 mL of 1% C_6_N_6_FeK_3_. The reaction mixture was incubated at 50 °C for 20 min. After the addition of 2.5 mL 10% C_2_HCl_3_O_2_, the mixture was centrifuged at 5000× *g* for 10 min. The supernatant was recovered, and we added 0.5 mL of 0.1% fresh FeCl_3_. Absorbance was recorded at 700 nm. Ascorbic acid was used as a positive control [60]. The percent increase in reducing power was calculated:$$\mathrm{Reducing}\;\mathrm{power}\;(\%)=[1-\frac{absorbance\;of\;control-absorbance\;of\;sample}{absorbance\;of\;control}]\times 100$$

A half maximal inhibitory concentration (IC_50_) value was calculated from the percentage reducing activity.

#### 2.9.2. Hydrogen Peroxide Scavenging Activity

The H_2_O_2_ scavenging activity was assessed by following the method of Bahorun et al. [61]. The mixture containing 1 mL of sample (0.1–2 mg mL^−1^), 2.4 mL of 0.1 M phosphate buffer (pH 7.4), and 0.6 mL of 40 mM H_2_O_2_ solution was shaken vigorously. After incubation for 10 min, absorbance was measured at 230 nm using ascorbic acid as a positive control. The H_2_O_2_ scavenging activity was calculated as follows:$$\mathrm{Scavenging}\;\mathrm{activity}\;(\%)=[1-\frac{absorbance\;of\;control-absorbance\;of\;sample}{absorbance\;of\;control}]\times 100$$

IC_50_ value was calculated from the percentage scavenging activity.

#### 2.9.3. Total Antioxidant Capacity

Total antioxidant activity (TAOC) was determined by the phosphomolybdenum method [62]. Extracts were prepared by taking 30 mg of samples in 5 mL ethanol. To 300 µL extract, 3 mL reaction mixture (0.6 M H_2_SO_4_, 28 mM Na_3_PO_4_, and 4 mM (NH_4_)_6_Mo_7_O_24_) was added, followed by incubation at 95 °C for 90 min. Absorbance was measured at 695 nm, and results were calculated as α-tocopherol equivalent (TE) µg g^−1^ DW and ascorbic acid equivalent (AAE) µg g^−1^ DW.

### 2.10. Nutrient Analyses

Dried mushrooms (250 mg) were digested with HNO_3_/HClO_4_/H_2_SO_4_ (4:2:0.5) in Kjeldahl tubes. Later, the mixture was allowed to cool, and the volume was made up to 50 mL by adding milliQ water after passing through Whatman No. 44 filter paper.

Total phosphorus was determined following the method described by Allen [63]. To a 3 mL aliquot of digested sample, 0.4 mL of 0.1% (NH_4_)_2_MoO_4_ prepared in 70% H_2_SO_4_ and 0.4 mL of 0.1% SnCl_2_ prepared in 2% HCl were added. The volume of the reaction mix was made up to 8 mL with distilled water. Following incubation in the dark for 30 min, absorbance was recorded at 715 nm. KH_2_PO_4_ was used for a standard calibration curve.

Elemental analysis of minerals: Cu, Mg, Ca, Fe, Al, and Zn were carried out by atomic absorption spectrometry. While C, N, and S were quantified by CHNS analyzer Vario EL cube (Elementar Analysensysteme GmbH, Langenselbold, Germany), Na and K were estimated using a Systronics flame photometer 128 (Systronics, New Delhi, India).

### 2.11. Antinutrients and Mineral Bioavailability

#### 2.11.1. Phytic Acid

For phytic acid estimation, 100 mg of each of the samples were extracted with 10 mL 2.4% HCl for 1 h and centrifuged at 3000× *g* for 20 min. In 3 mL of supernatant, 1 mL of Wade reagent was added, and optical density at 500 nm was recorded. A series of calibration standards were prepared from sodium salt of phytic acid [64]. The results were expressed as mg phytic acid equivalent g^−1^ (mg PAE g^−1^ DW).

#### 2.11.2. Condensed Tannins

Condensed tannins were determined according to Price and Butler [65] by taking 50 µL extract in 1.5 mL of 4% vanillin and 750 µL of HCl. Following incubation for 20 min, absorbance was recorded at 500 nm. The results were expressed as mg tannic acid equivalent g^−1^ DW (mg TAE g^−1^ DW).

#### 2.11.3. Molar Ratios of Antinutrients to Minerals

The bioavailabilities of minerals were estimated by calculating antinutrients to minerals molar ratios [66].

### 2.12. Vitamins

#### 2.12.1. Vitamin C

The method described by Nielsen was used to quantify vitamin C concentration [67]. Sample was extracted with 1% HPO_3_ for 45 min at room temperature and filtered with Whatman No.1 filter paper. In 0.5 mL filtrate, 4.5 mL 0.1% DCPIP was added. Absorbance was measured within 30 min at 515 nm. Ascorbic acid was used for standard calibration curve.

#### 2.12.2. Vitamin B_3_ and B_6_

Estimation of vitamin B_3_ and B_6_ was performed by taking 1 mL sample, 3 mL of 0.2 M KI, and 0.5 mL ethanol. After mixing thoroughly and incubating for 15 min, absorbance was recorded at 290 nm. Niacin and pyridoxine were used as standard for calibration of vitamins B_3_ and B_6_, respectively [68].

### 2.13. Statistical Analysis

All the analyses were performed in triplicate, and the results are expressed as mean ± standard deviation (SD). Statistical analyses were carried out with SPSS software version 21.0 (IBM Corporation, New York, United States). Student’s *t*-test was used to compare the differences between the two mushrooms. Significant differences were considered at the level of *p* < 0.05.

## 3. Results

### 3.1. Identification of Mushrooms

Species 1 was identified as *Lactarius drassinus* (Figure 1A,B), and species 2 as *Lactarius controversus* (Figure 1C,D), according to their macromorphological features and responses to different chemical tests, confirmed according to Verma et al. [46] and Pekşen et al. [41]. Furthermore, molecular identification results showed sequence similarity (≥98%) against the reference sequences from GenBank and were confirmed as *L. drassinus* and *L. controversus*. Subsequently, FASTA sequences were uploaded on NCBI, and the accession numbers are OM680959 and OM680932, respectively. The phylogenetic trees were constructed using Mega software version 11.0.10 [69] to confirm the evolutionary relationship amongst the *Lactarius* spp. (Appendix A).

### 3.2. GC-MS-Based Metabolite Profiling

GC-MS analysis resulted in the identification of 88 compounds each in *L. drassinus* and *L. controversus* (Table 1). The identified compounds included amino acids, organic acids, fatty acids, sugars, and their derivatives, among others. Of the total 132 identified compounds, forty-four compounds, namely, 5-oxoproline, serine, valine, phenylalanine, threonine, isoleucine, leucine, citramalic acid, fumaric acid, isocitric acid, ketoisocaproic acid, malic acid, oxalic acid, phosphoric acid, succinic acid, phosphoric acid, arachidic acid, myristic acid, palmitic acid, stearic acid, linoelaidic acid, androstenone, ergosterol, trehalose, fructofuranose, fructose, galactopyranose, lactose, ribofuranose, adonitol, arabinitol, glucitol, erythritol, maltitol, myo-inositol, glycerol, gluconic acid, threonic acid, N-acetyl glucosamine, anthraergostatetraenol benzoate, globulol, mansonone, 2-ethylhexyl acetate, and desacetylcinobufotalin, were common in both the mushrooms studied.

The most abundant class of compounds was sugars and their derivatives, wherein a total of 48 compounds contributing to 42.8% and 33.45% of the total composition in *L. drassinus* and *L. controversus*, respectively, were identified. Sugar alcohols were the major sugars found, with the most abundant constituent being glucitol (17.39% and 20.44%) in *L. drassinus* and *L. controversus*, respectively. Non-structural carbohydrates and sugar acids accounted for 10.22% and 1.04% in *L. drassinus*, and 4.73% and 0.62% in *L. controversus*, respectively. Other sugar derivatives contributed to a meagre 0.54% in *L. controversus*. On the contrary, *L. drassinus* showed distinctive content of sugar-derivative N-acetyl glucosamine.

The second major class was fatty acids and their derivatives, accounting for 10.75% and 32.08% in *L. drassinus* and *L. controversus*, respectively. A total of 32 fatty acids and their derivatives were present in both the species. Unsaturated fatty acids were found to be 19.3% in *L. controversus*, wherein oleic acid and linoleic acid were predominant. Contrarily, unsaturated fatty acids constituted only 0.60% in *L. drassinus*. The saturated fatty acids in *L. drassinus* and *L. controversus* were 8.15% and 9.62%, respectively. The major contributor was stearic acid in both the species. It was observed that both the mushrooms were also rich in sterols, with ergosterol being the predominant sterol.

Organic acid was the third most representative class of compounds, and its concentrations were 18.42%, and 9.99% in *L. drassinus* and *L. controversus*, respectively. Malic acid was the most abundantly present organic acid. Apart from organic acids, inorganic acids such as borinic acid and phosphoric acid were also present.

Amino acid profile showed a great variation in terms of percent composition, constituting 11.71% and 1.66% in *L. drassinus* and *L. controversus*, respectively. Overall, six essential and ten non-essential amino acids were detected in the studied mushrooms. Of all the amino acids, 5-oxoproline was found to be predominant in both the mushrooms.

### 3.3. Proteins

The concentrations of total protein were 293.33 and 337.78 mg g^−1^ DW in *L. drassinus* and *L. controversus*, respectively (Figure 2A).

### 3.4. Carbohydrates and Sugars

Between the two mushrooms, *L*. *drassinus* recorded higher concentrations of TAC and TSS than *L. controversus* (Figure 2B,C). Similarly, *L. drassinus* was found to contain RS and NRS in higher concentrations than *L. controversus* (Figure 2D,E). On the contrary, starch concentrations were found higher in *L. controversus* than *L. drassinus* (Figure 2F).

### 3.5. Total Phenol and Total Flavonoid

The estimation of total phenol revealed that *L. controversus* contained 104.53 mg GAE g^−1^ DW of total phenol and 99.38 mg GAE g^−1^ DW in *L. drassinus* (Figure 3A). Similar results were obtained for total flavonoids, where 84.39 mg QE g^−1^ DW was present in *L. controversus* and 72.27 mg QE g^−1^ DW in *L. drassinus* (Figure 3B).

### 3.6. UPLC-Based Quantification of Phenolic Compounds

The profile of phenolic compounds is given in Table 2. Chlorogenic acid was found highest in concentration in both the mushrooms, 54.27 µg g^−1^ DW in *L. drassinus* and 30.24 µg g^−1^ DW in *L. controversus*. The concentration of cinnamic acid was least in *L. drassinus*; however, the same was not detected in *L. controversus*.

### 3.7. Carotenoids

Lycopene was higher in *L. drassinus* than *L. controversus*. β-Carotene, however, was found in *L. controversus* in a higher concentration than that of *L. drassinus* (Table 3).

### 3.8. Antioxidant Potential

The assay of H_2_O_2_ scavenging activity showed low IC_50_ values in both *Lactarius* spp., corresponding to high scavenging activities (Figure 4A). Similarly, reducing power with low IC_50_ values was observed in *L. drassinus* and *L. controversus* (Figure 4B).

### 3.9. Total Antioxidant Capacities

*L. drassinus* showed a higher concentration of fat-soluble antioxidants than in *L. controversus*. A similar trend was observed in water-soluble antioxidants (Table 3).

### 3.10. Minerals

Of the seven macroelements estimated, C, P, and Ca were found to be higher in *L. drassinus* than *L. controversus*, and vice versa for N, K, S, and Mg (Table 4). Conversely, in that of microelements, *L. drassinus* accumulated more Cu, Na, and Fe than *L. controversus*. Al, Ni, and Zn were observed to be higher in *L. controversus* (Table 5).

### 3.11. Antinutrients and Mineral Bioavailability

Condensed tannin concentration was 0.164 mg TAE g^−1^ DW in *L. drassinus* and 0.156 mg TAE g^−1^ DW in *L. controversus* (Figure 3C). In estimation of phytic acid, *L. controversus* showed a higher concentration (0.268 mg PAE g^−1^ DW) than *L. drassinus* (0.246 mg PAE g^−1^ DW) (Figure 3D). Oxalic acid concentrations were 0.94% and 0.01% in *L. drassinus* and *L. controversus*, respectively (Table 1).

Molar ratios of antinutrients to minerals were calculated (Table 6). [Phytate]:[Fe], [Phytate]:[Ca], and [Phytate]:[Zn] were lower in *L. drassinus* than *L. controversus*. However, no significant difference was observed in [Phytate]:[Ca] and [Phytate]:[Zn] in the two mushrooms. [Oxalate]:[Ca + Mg] was found to be low.

### 3.12. Vitamins

Analysis for vitamins revealed that concentrations of vitamins C, B_3_, and B_6_ were higher in *L. drassinus* than *L. controversus* (Table 3).

## 4. Discussion

Around the world, macrofungi are relished for their flavour, as well as nutritional and medicinal advantages. A great deal of work has been aggressively executed in this domain worldwide; however, nutritional investigation of mushrooms from cold desert areas, such as Kargil, has not been accomplished. The present study characterised the nutritional and nutraceutical potential of two undocumented wild edible mushrooms, *L. drassinus* and *L. controversus*, from Kargil.

Carbohydrates play a crucial role in the body by providing and storing energy, building macromolecules, and preserving protein and fat for alternative purposes [72]. In contrast with previous studies on wild edible mushrooms [73,74], TAC were found in higher range in both the species studied. GC-MS analysis revealed that the *Lactarius* spp. are rich in sugars, wherein their alcoholic derivatives such as glucitol, glycerol, arabitol, and erythritol are predominantly present. Owing to their well-recognised role in combating stress by redox homeostasis, a predominant presence of sugar alcohols could be perceived as a stress tolerance strategy adopted by the mushrooms to withstand the constant pressures of drought and low temperature [75]. Pharmaceutically, sugar alcohols are used as a sugar alternative in diabetic foods and as an effective dietary approach as they contain a low number of calories and possess non-fermenting properties [76,77]. Moreover, high sugar composition in GC-MS profile was further supported by TSS estimated spectrophotometrically. The TSS in the two *Lactarius* spp. were found to be higher than those reported for *Lentinus* spp., *Schizophyllum* sp., and *Termitomyces* sp. [78]. RS levels, which are indicative of food quality, were found to be high. Similarly, NRS has been reported to increase under drought stress [79], which explains their high accumulation in the studied mushrooms. In contrast to sugars and their alcoholic derivatives, starch concentration was found to be lower, making these macrofungi a healthy choice for dietary management of diabetes mellitus, as starch can potentially interfere with blood glucose levels [80].

Alongside fats and carbohydrates, proteins are one of the three principal macronutrients in the human diet, and their deficiency is one of the world’s most acute nutritional problems [81]. In this regard, mushrooms offer a protein-rich diet that could be a promising remedy against protein malnutrition problems. Mushroom proteins are reportedly known for their immunomodulatory properties and are thus considered to be a new class of bioactive proteins that have potential use as an adjuvant for tumour therapy [9]. In this context, *L. drassinus* and *L. controversus* can be exploited as good sources of proteins as they exhibit higher concentration (almost twice) than previously reported values in *L. controversus* from the Middle Black Sea Region of Turkey and *L. deliciosus* from Macedonia, as well as other edible mushrooms including *Agaricus bisporus*, *Flammulina velutipes*, *Letinus edodes*, and *P. eryngii*, among others [82,83].

The presence of amino acids, particularly essential amino acids, is credited for the superior quality of mushroom proteins [84]. In comparison, *L. drassinus* was found to be richer in both essential and non-essential amino acids than *L. controversus*. Amongst the amino acids identified, the non-essential amino acid 5-oxoproline was detected in the highest concentration. 5-Oxoproline plays an important role in forming the characteristic umami taste in mushrooms [85]. Moreover, it possesses several bioactivities such as antibacterial, antioxidant, antidiabetic, and anti-inflammatory activities [86], and it has been recently reported to be a potent antiviral candidate in the treatment of COVID-19, as well as Alzheimer’s disease [87,88]. Other main amino acid constituents detected in *L. drassinus* were aspartic acid, glutamic acid, and lysine. Higher aspartic and glutamic acid concentrations correlated to the fact that they serve as the precursors from which the backbones of other amino acids are formed [89]. Lysine, on the contrary, is an essential amino acid, generally found to be limited in most vegetable proteins. It was present in an appreciable amount in the examined mushrooms, analogous to reports from other studies [90,91]. However, it is noteworthy that these compounds were not detected in *L. controversus*. Apart from their primary role as building blocks of proteins, amino acids are also known for their antioxidant potency [92]. Therefore, a significant composition of the amino acids can be linked to the tolerance mechanism adopted by these mushrooms in response to cold stress [93,94].

Mushrooms have the potential to contribute immensely to the provision of both macro- and micronutrients. Elemental analyses revealed the presence of Ca, Mg, C, N, S, Zn, Ni, Cu, K, and Al in moderate to high concentrations [95,96]. Higher accumulation of mineral elements such as Cu, Fe, Zn, Ca, and Mg in the studied mushrooms insists that they should be included in the diet to combat mineral deficiencies. Although P and Na concentrations were in concordance with previous reports, mushrooms are generally considered to be low in the two minerals [96,97]. Low Na and high K in the investigated mushrooms indicates the importance of incorporating these mushrooms into the diet for impeding hypertension [98]. The mineral values of these mushrooms can guarantee their healthy choice as supplementary foods to the population of Kargil and other states that predominantly rely on a cereal diet for mineral requirement.

It is notable that the two mushrooms investigated are not only rich in minerals, but also in terms of their bioavailabilities, which is attributable to low concentration of antinutrients, namely, phytate, tannins, and oxalate. Phytate forms stable complexes with mineral ions, such as Ca, Mg, Zn, and Fe, resulting in the formation of insoluble salts, which renders them unavailable for uptake in the intestines [99,100]. Oberleas and Harland reported that foods having a molar ratio of [Phytate]/[Zn] less than 10 showed substantial availability of Zn [70]. Likewise, Hassan et al. reported [Phytate]/[Ca] and [Phytate]/[Fe] below 0.2 and 0.4, respectively, indicating adequate bioavailabilities of Ca and Fe [71]. In this context, the estimated [Phytate]/[Fe], [Phytate]/[Ca], and [Phytate]/[Zn] in the tested mushrooms were lower than their critical values, indicating better bioavailabilities of Fe, Ca, and Zn. Considering the phytate content of mushrooms generally lower than that of green leafy vegetables [101], mushrooms could be recommended as a superior food to combat nutrient deficiencies.

Tannins are one of the most studied antinutrients, whose composition above 10% of the total DW interferes with protein degradation and digestibility [102,103]. Tannins are also known as potent antioxidants [104]. In the studied samples, although tannins were present in low concentrations compared with previously reported values in *Lactarius* spp. [105,106], the higher antioxidant activities could have been due to the presence of other antioxidants such as phenols and flavonoids. Another antinutrient, oxalate, forms insoluble salts by binding to minerals, thereby increasing the likelihood of kidney stone formation [107]. Interestingly, oxalates in the studied mushrooms were quite low for *L*. *controversus* when compared to the content reported in spinach (1.14%) and almonds (0.47%) [108]. Biological interaction of Ca, Mg, and oxalate were found lower than the critical value (2.5) in the studied mushrooms [70], indicating high availabilities of Ca and Mg. It can be deduced that in both the *Lactarius* spp., the concentrations of phytates, tannins, and oxalates were insignificant in terms of interfering with mineral absorption and digestibility of proteins, and hence they should not affect their nutritional potential.

Fatty acids have varied functionalities in cells, ranging from structural “building blocks” of plasma membranes to facilitators of energy and signalling compounds [109]. The assessment of saturated fatty acids of the two mushrooms diagnosed stearic acid and palmitic acid as the major contributor. Similar constituents of fatty acid have been reported in various mushrooms, given that the concentration differs from species to species [110,111]. High accumulation of unsaturated fatty acid is a feature of stress acclimation as they are known to lower phase-transition temperature and thereby assist in adapting to low temperature [112]. Interestingly, though saturated fatty acid concentration was similar, that of unsaturated fatty acids varied between the two mushrooms. While only two unsaturated fatty acids, methyl linoleate and linoelaidic acid, were detected in *L. drassinus*, *L. controversus* was found to be rich in unsaturated fatty acids, especially linoleic acid (omega-6) and oleic acid (omega-9). In a clinical study by Richard et al., omega-3 and omega-6 fatty acids were reported to possess antioxidant activity in vascular endothelial cells, thereby subsiding inflammation and eventually reducing the risk of disease occurrence such as atherosclerosis and cardiovascular diseases [113]. Moreover, the composition of unsaturated fatty acids was observed to be higher than that of saturated fatty acid in *L. controversus*, testifying it as a legit source of healthy fats.

Ergosterol, a fatty acid derivative and fungal alternative to cholesterol, is crucial in regulating permeability and fluidity of fungal membranes [114]. It was found to be present in higher concentration in comparison with other previous reports in other edible mushrooms (0.2–7.8 mg g^−1^) [115,116]. Ergosterol is also a biological precursor of vitamin D_2_, making these mushrooms a reserve to meet vitamin D intake for vegetarians, besides dairy and its products [117]. Additionally, ergosterols are recognised for antitumour, anti-inflammatory, and antioxidant activities, as well as decreasing COVID-19 implications [118]. The potential of mushrooms for these reasons needs to be sufficiently appreciated.

Deficiencies of vitamins represent a serious health issue on the global food map. On that note, the vitamin profile indicated higher concentrations of vitamins B_3_ and B_6_ and tocopherols than previous findings in wild edible mushrooms [119,120,121]. Tocopherols, collectively called vitamin E, are fat-soluble vitamins best known for their antioxidant activities. Moreover, they are associated with cancer-preventive activities [122]. Significant concentrations of vitamins in the mushrooms can, hence, be vouched for in redemption of vitamin deficiencies.

Numerous organic and mineral acids are present in the mushrooms studied, with malic acid, pyruvic acid, and fumaric acid as chief components. Besides primary metabolic activities, they are reported to be potent antimicrobial agents [123]. Other functions include prevention of rancidity of fats and oils due to their synergistic effect with antioxidants, as well as being food acidity regulators and flavour enhancers [124]. In the case of mineral acids, phosphoric acid was detected at a high concentration. It is often used in the food and beverage industries as an additive and acidulant. In view of pharmaceutical functions, it is known for lowering blood pH and has been used in the treatment of lead poisoning [124].

The investigated mushrooms showed high concentrations of total phenol and flavonoids, comparatively higher than previously reported values in other wild mushrooms [125]. The bioactivities of phenolics can be related to their metal chelating ability, lipoxygenase inhibition, and free radical scavenging capacity [126]. Flavonoids are known to scavenge free radicals and terminate chain reactions taking place during the oxidation of triglycerides that reportedly play a protective role in diseases linked to oxidative stress, such as cardiovascular diseases and cancer [127]. Congruent with studies on other edible mushrooms, phenols were found to be the major antioxidant in *L. drassinus* and *L. controversus*, followed by flavonoids [128]. The profile of phenolic compounds puts emphasis on chlorogenic acid, which is an important dietary polyphenol playing several roles of therapeutic importance [129]. In addition to its role as an antioxidant, chlorogenic acid has been reported to possess several bioactivities including antimicrobial, hepatoprotective, cardioprotective, anti-inflammatory, antipyretic, anti-obesity, and anticancer activities [130]. Kalogeropoulos et al. [131] and Heleno et al. [120] reported lower concentrations of gallic acid, vanillin, ferulic acid, cinnamic acid, and quercetin dihydrate in wild edible mushrooms from the island of Lesvos (Greece) and Northeast Portugal in comparison to the mushrooms from Kargil in the present study. Thus, these mushrooms can be inferred as potential candidates for food supplementation, owing to their high phenolic contents. Furthermore, lycopene and β-carotene concentrations were estimated. They are the predominant carotenoids found in various mushrooms, both wild and cultivated [132]. β-Carotene functions as antioxidants as well as provitamin A, and it plays key role as one of the vitamin A dietary sources [133]. On the other hand, lycopene is a precursor for β-carotene synthesis and is known to be a potent antioxidant because of its unsaturated nature [132].

A quantitative measure to correlate antioxidants with antioxidant potency was studied. As depicted by lower IC_50_ values in assays of H_2_O_2_ scavenging activity and reducing power, both *Lactarius* spp. correspond to high scavenging activities and enhanced reducing ability. Free radical scavengers inhibit oxidative lipid degradation, which otherwise can be deleterious to the cellular components and their functions [134]. With the radical scavenging activity mediated by phenols, flavonoids, and carotenoids, these wild edible mushrooms can be recommended as natural healthy antioxidants.

## 5. Conclusions

This study documents the metabolite profiles of two wild edible mushrooms, *L. drassinus* and *L. controversus*, collected from a unique habitat consisting of cold desert terrains in Kargil. The two mushrooms are rich in proteins, carbohydrates, vitamins, and several minerals, and they are low in antinutritional compounds, such as condensed tannins, oxalic acid, and phytic acid. They exhibit effective antioxidant properties by virtue of high concentrations of phenols, flavonoids, and carotenoids. In comparison, *L. drassinus* exhibited a higher concentration of phenolic acids (gallic acid, chlorogenic acid, vanillin, ferulic acid, and cinnamic acid) and flavonoids (quercetin dihydrate) than *L. controversus*. On the contrary, *L. controversus* showed higher carotenoid concentration than *L. drassinus*. Moreover, metabolite profiling revealed the presence of compounds of pharmaceutical importance such as 5-oxoproline, glucitol, and ergosterol. Thus, *L. drassinus* and *L. controversus* offer well-balanced sources of nutritional and nutraceutical compounds, and it is imperative to communicate the nutritional database of mushrooms found in hostile regions, such as the cold terrains of Kargil, where acute shortage of food, and hence nutrients, has always been a major challenge to the population. Moreover, future studies should be directed towards cultivation of these mushrooms so that they can be made available to the local population of cold desert as a dietary resource.

## Figures and Tables

**Figure 1 jof-09-00763-f001:**
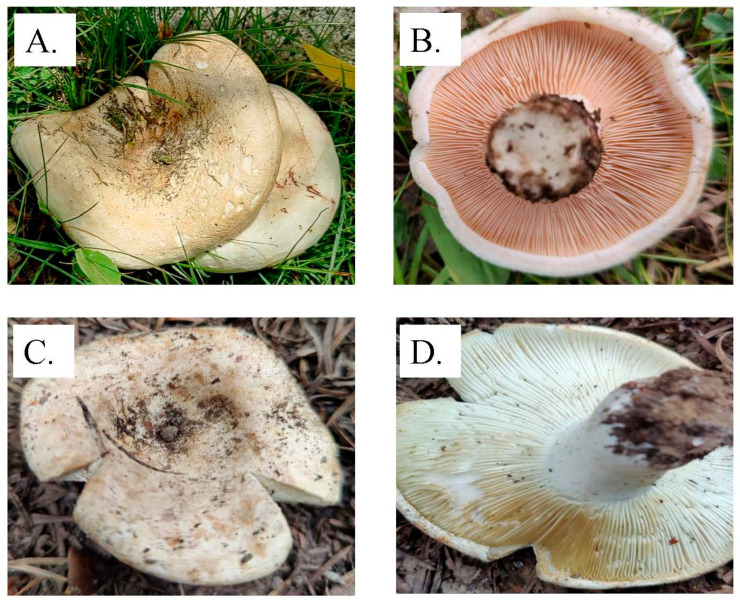
Basidiomata of *L. drassinus* (**A**,**B**) and *L. controversus* (**C**,**D**).

**Figure 2 jof-09-00763-f002:**
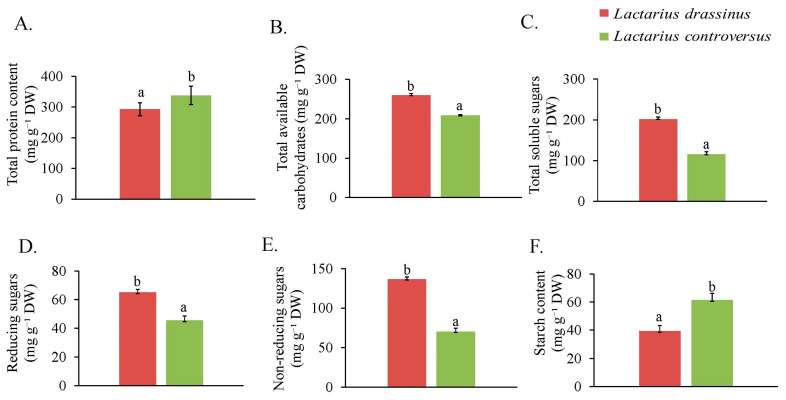
Concentration of total protein content (**A**), total available carbohydrates (**B**), total soluble sugars (**C**), reducing sugars (**D**), non-reducing sugars (**E**), and starch content (**F**) in *L. drassinus* and *L. controversus*. Values represent mean ± SD (*n* = 3). Different letters within a row represent significant difference at *p* ≤ 0.05, derived from Student’s *t*-test.

**Figure 3 jof-09-00763-f003:**
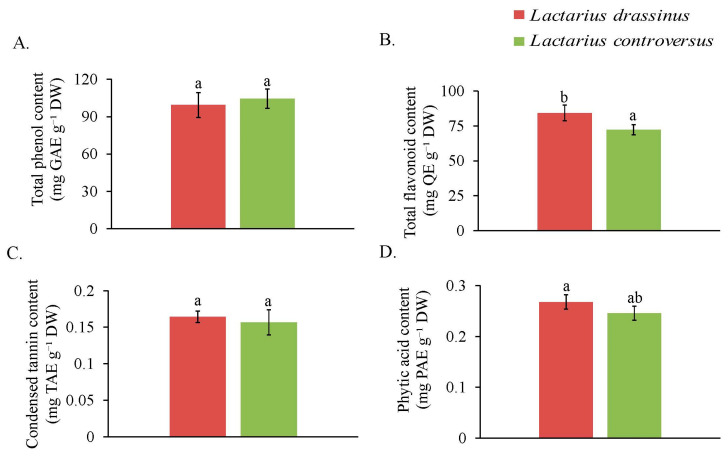
Content of antioxidants: total phenol (**A**) and total flavonoid (**B**), and concentration of antinutrients: condensed tannins (**C**) and phytic acid (**D**) in *L. drassinus*, and *L. controversus*. Values represent mean ± SD (*n* = 3). Different letters within a row represent significant difference at *p* ≤ 0.05, derived from Student’s *t*-test.

**Figure 4 jof-09-00763-f004:**
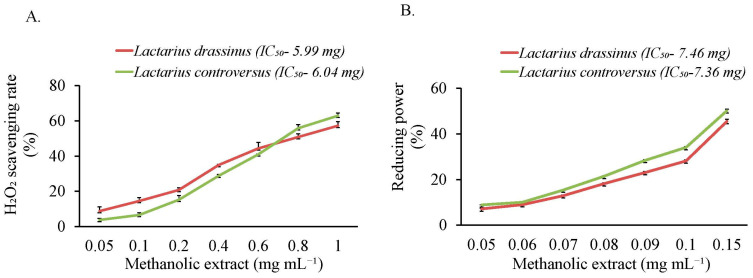
Hydrogen peroxide scavenging activity (**A**) and reducing power (**B**) of *L. drassinus* and *L. controversus*. Values represent mean ± SD (*n* = 3).

**Table 1 jof-09-00763-t001:** Compounds identified by gas-chromatography–mass-spectrometry-based metabolite profiling of *L. drassinus* and *L. controversus* and their percent composition.

Class	Compound Name	*L. drassinus*	*L. controversus*
Amino acids and their derivatives			
Non-essential amino acids	Alanine	0.48 ± 0.08	nd
	Glycine	nd	0.03 ± 0
	Homocysteine	0.54 ± 0.18	nd
	5-Oxoproline	5.21 ± 1.98 ^b^	0.70 ± 0.15 ^a^
	β-Alanine	0.11 ± 0.03	nd
	Proline	0.37 ± 0.07	nd
	Serine	0.66 ± 0.27 ^b^	0.28 ± 0.10 ^a^
	Asparagine	0.55 ± 0.21	nd
	Aspartic acid	0.93 ± 0.38	nd
	Glutamic acid	0.82 ± 0.21	nd
Essential amino acids	Valine	0.50 ± 0.10 ^b^	0.07 ± 0.02 ^a^
	Phenylalanine	0.37 ± 0.23 ^b^	0.03 ± 0.01 ^a^
	Threonine	0.58 ± 0.23 ^b^	0.06 ± 0 ^a^
	Isoleucine	0.27 ± 0.03 ^a^	0.24 ± 0.05 ^a^
	Leucine	0.34 ± 0.10 ^ab^	0.20 ± 0.07 ^a^
	Lysine	0.91 ± 0.21	nd
Acids and their derivatives			
Organic acids and their derivatives	Butylethylmalonic acid	nd	0.48 ± 0.10
	Citramalic acid	0.25 ± 0.13 ^b^	0.04 ± 0.03 ^a^
	Dimethylolpropionic acid	0.24 ± 0.05	nd
	Fumaric acid	3.25 ± 0.69 ^b^	1.60 ± 0.68 ^a^
	Hydroxycaproic acid	0.37 ± 0.02	nd
	Isocitric acid	0.58 ± 0.24 ^a^	2.03 ± 0.71 ^b^
	Itaconic acid	nd	0.48 ± 0.15
	Ketoisocaproic acid	0.24 ± 0.17 ^b^	0.07 ± 0.01 ^a^
	Maleic acid	0.95 ± 0.07	nd
	Malic acid	9.97 ± 5.71 ^ab^	3.14 ± 1.29 ^a^
	Oxalic acid	0.94 ± 0.10 ^b^	0.01 ± 0 ^a^
	Pentonic acid	0.24 ± 0.06	nd
	Succinic acid	2.11 ± 0.72 ^b^	1.10 ± 0.8 ^a^
	Dibromo-succinic acid	nd	0.53 ± 0.37
	Isobutyl phthalate	nd	0.51 ± 0.23
	Traumatic acid	0.22 ± 0.04	nd
Mineral acids	Borinic acid	0.24 ± 0.07	nd
	Phosphoric acid	1.65 ± 0.50 ^a^	3.92 ± 0.56 ^b^
Fatty acids and their derivatives			
Saturated fatty acids	Arachidic acid	0.27 ± 0.15 ^b^	0.11 ± 0.01 ^a^
	Butyric acid	nd	0.10 ± 0.06
	Margaric acid	nd	0.40 ± 0.21
	Methyl stearate	nd	1.21 ± 0.46
	Myristic acid	0.87 ± 0.08 ^a^	0.23 ± 0.03 ^b^
	Nonanoic acid	nd	0.12 ± 0.08
	Palmitic Acid	2.93 ± 0.06 ^b^	1.89 ± 0.33 ^a^
	Prostadienoic acid	0.16 ± 0.03	nd
	Stearic acid	4.19 ± 0.70 ^a^	5.77 ± 2.19 ^a^
Unsaturated fatty acids	Linoelaidic acid	0.25 ± 0.04 ^a^	0.16 ± 0.05 ^a^
	Linoleic acid	nd	5.82 ± 1.99
	Methyl linoleate	0.35 ± 0.20	nd
	Monolinolein	nd	0.47 ± 0.23
	Myristoleic acid	nd	0.21 ± 0.09
	Norlinolenic acid	nd	0.28 ± 0.02
	Oleic acid	nd	11.23 ± 6.21
	Oleoylglycerol	nd	0.92 ± 0.37
	Palmitelaidic acid	nd	0.21 ± 0.03
Other fatty acid derivatives	Alpha dihydrotestosterone	nd	0.61 ± 0.25
	Androstenone	0.45 ± 0.11 ^a^	0.61 ± 0.14 ^ab^
	Campesterol	nd	0.05 ± 0
	Ergostatetraenol	0.15 ± 0.08	nd
	Ergosterol	0.23 ± 0.06 ^a^	0.86 ± 0.21 ^b^
	Hydroxyandrosterone	0.21 ± 0.03	nd
	Hydroxydehydroandrosterone	nd	0.29 ± 0.14
	Methandrostenolone	nd	0.41 ± 0.11
	Methoxyestradiol	0.34 ± 0.08	nd
	Prostaglandin	0.21 ± 0.08	nd
	Stigmasterol	nd	0.06 ± 0
	Tetrahydrocortisol	nd	0.06 ± 0.02
	Tricaprin	0.28 ± 0.05	nd
	Monocaprin	0.13 ± 0.03	nd
Sugars and their derivatives			
Sugars	Allofuranose	nd	0.15 ± 0.06
	Cellobiose	0.84 ± 0.17	nd
	Deoxyglucose	0.05 ± 0.01	nd
	Deoxyribose	nd	0.83 ± 0.18
	Psicofuranose	0.15 ± 0.03	nd
	Trehalose	0.04 ± 0.02 ^a^	0.07 ± 0.04 ^b^
	Erythrotetrofuranose	0.22 ± 0.03	nd
	Fructofuranose	0.20 ± 0.02 ^a^	0.37 ± 0.11 ^b^
	Fructopyranose	nd	0.12 ± 0.08
	Fructose	1.69 ± 0.37 ^b^	0.53 ± 0.12 ^a^
	Galactopyranose	0.12 ± 0.03 ^a^	0.17 ± 0.08 ^a^
	Lactose	0.13 ± 0.04 ^ab^	0.09 ± 0.03 ^a^
	Maltose	0.23 ± 0.17	nd
	Mannose	0.21 ± 0.04	nd
	Melibiose	0.21 ± 0.07	nd
	Ribofuranose	0.11 ± 0.03 ^a^	0.38 ± 0.06 ^b^
	Sucrose	4.00 ± 0.19 ^b^	2.02 ± 0.81 ^a^
	Talose	2.02 ± 0.11	nd
Sugar alcohols	Adonitol	0.77 ± 0.22 ^ab^	0.49 ± 0.17 ^a^
	Arabinitol	0.13 ± 0.07 ^a^	0.19 ± 0.05 ^a^
	Arabitol	nd	3.29 ± 1.43
	Deoxyribitol	0.07 ± 0.02	nd
	D-Glucitol	17.39 ± 7.14 ^a^	20.44 ± 5.51 ^a^
	Erythritol	0.28 ± 0.05 ^a^	2.57 ± 1.02 ^b^
	Fucitol	nd	0.28 ± 0.09
	Maltitol	0.82 ± 0.23 ^b^	0.05 ± 0.02 ^a^
	Myo-inositol	2.04 ± 0.44 ^b^	0.07 ± 0.03 ^a^
	Xylitol	nd	0.10 ± 0.01
	Glycerol	4.64 ± 1.57 ^b^	0.08 ± 0.03 ^a^
Sugar acids	Arabinonic acid	0.03 ± 0.01	nd
	Galactonic acid	nd	0.25 ± 0.16
	Gluconic acid	0.48 ± 0.08 ^b^	0.15 ± 0.07 ^a^
	Glucuronic acid	0.15 ± 0.02	nd
	Threonic acid	0.38 ± 0.09 ^b^	0.22 ± 0.06 ^a^
Other sugar derivatives	Butanetriol	nd	0.02 ± 0.01
	Glucosamine	0.41 ± 0.07	nd
	Acetin	nd	0.45 ± 0.20
	N-Acetyl glucosamine	4.90 ± 1.12 ^b^	0.06 ± 0.03 ^a^
	Sorbitol phosphate	0.09 ± 0.03	nd
	Gluconolactone	nd	0.07 ± 0.02
Other compounds	Anthraergostatetraenol benzoate	0.19 ± 0.09 ^a^	0.25 ± 0.11 ^a^
	Costunolide	nd	0.53 ± 0.03
	Furosardonin A	nd	1.96 ± 0.35
	Globulol	3.70 ± 0.11 ^a^	4.52 ± 2.19 ^ab^
	Lavandulol	nd	1.21 ± 0.06
	Mansonone	0.03 ± 0.01 ^a^	1.35 ± 1.01 ^b^
	Santamarine	nd	0.52 ± 0.07
	Vellerdiol	nd	0.13 ± 0.05
	Naphthalenediol	0.15 ± 0.04	nd
	Stahlianthusone	nd	0.25 ± 0.13
	Acetoin	nd	0.21 ± 0.09
	Chrysorrhedial	nd	0.14 ± 0.06
	2-Ethylhexyl acetate	0.08 ± 0.02 ^a^	0.06 ± 0.03 ^a^
	Rhizoxin	nd	0.17 ± 0.04
	Methyl 2-hydroxytricosanoate	0.38 ± 0.08	nd
	Furanether A	nd	3.44 ± 0.59
	Methoxypropene	0.50 ± 0.26	nd
	Methyladenosine	0.16 ± 0.02	nd
	Adenosine	0.23 ± 0.03	nd
	Ammelide	0.56 ± 0.11	nd
	Methyluridine	1.12 ± 0.14	nd
	Niacinamide	0.06 ± 0.02	nd
	Ethanolamine	nd	0.08 ± 0
	Uracil	0.11 ± 0.02	nd
	Uridine	nd	0.52 ± 0.18
	Desacetylcinobufotalin	0.03 ± 0.01 ^a^	0.37 ± 0.10 ^b^

nd—not detected; values represent mean ± SD (*n* = 3). Different letters within a row represent significant difference at *p* ≤ 0.05, derived from Student’s *t*-test.

**Table 2 jof-09-00763-t002:** Ultra-performance-liquid-chromatography-based quantification of phenolic compounds (expressed in µg g^−1^ DW) of *L. drassinus* and *L. controversus*.

Phenolic Compounds	*L. drassinus*	*L. controversus*
Gallic acid	13.05 ± 0.09 ^b^	12.35 ± 0.40 ^a^
Chlorogenic acid	54.27 ± 0.84 ^b^	30.24 ± 0.49 ^a^
Vanillin	14.22 ± 0.12 ^b^	10.49 ± 0.16 ^a^
Ferulic acid	17.83 ± 0.11 ^b^	12.23 ± 0.30 ^a^
Quercetin dihydrate	22.44 ± 0.64 ^b^	10.71 ± 0.88 ^a^
Cinnamic acid	0.52 ± 0.04	nd

nd—not detected; values represent mean ± SD (*n* = 3). Different letters within a row represent significant difference at *p* ≤ 0.05, derived from Student’s *t*-test.

**Table 3 jof-09-00763-t003:** Concentrations of vitamins, carotenoids, and antioxidant potential of *L. drassinus* and *L. controversus*.

Parameters	*L. drassinus*	*L. controversus*
Vitamins (µg g^−1^ DW)		
Vitamin C	37.34 ± 2.03 ^b^	27.18 ± 2.50 ^a^
Vitamin B_3_	122.33 ± 10.0 ^b^	98.67 ± 9.0 ^a^
Vitamin B_6_	12.76 ± 4.1 ^a^	11.28 ± 2.02 ^a^
Carotenoids (µg g^−1^ DW)		
β-Carotene	12.7 ± 4.55 ^b^	6.9 ± 0.77 ^a^
Lycopene	82.3 ± 9.6 ^a^	108.01 ± 11.44 ^b^
Antioxidant potential		
Total antioxidant (µg AAE g^−1^ DW)	34.34 ± 5.71 ^a^	28.9 ± 4 ^a^
Total antioxidant (µg TE g^−1^ DW)	308.09 ± 15.73 ^b^	246.33 ± 8.13 ^a^

Values represent mean ± SD (*n* = 3). Different letters within a row represent significant difference at *p* ≤ 0.05, derived from Student’s *t*-test.

**Table 4 jof-09-00763-t004:** Macroelement concentrations (expressed in mg g^−1^ DW) of *L. drassinus* and *L. controversus*.

Macroelements	*L. drassinus*	*L. controversus*
C	435.13 ± 0.58 ^b^	396.20 ± 0.29 ^a^
N	30.60 ± 0.26 ^a^	35.30 ± 0.29 ^b^
S	3.57 ± 0.08 ^a^	4.64 ± 0.12 ^b^
P	0.41 ± 0 ^b^	0.20 ± 0 ^a^
Mg	0.25 ± 0.01 ^a^	0.28 ± 0 ^b^
Ca	0.70 ± 0.07 ^b^	0.56 ± 0.02 ^a^
K	13.84 ± 0.30 ^a^	16.14 ± 0.55 ^b^

Values represent mean ± SD (n = 3). Different letters within a row represent significant difference at *p* ≤ 0.05, derived from Student’s *t*-test.

**Table 5 jof-09-00763-t005:** Microelement concentrations (expressed in µg g^−1^ DW) of *L. drassinus* and *L. controversus*.

Microelements	*L. drassinus*	*L. controversus*
Cu	14.02 ± 0.89 ^b^	11.59 ± 0.83 ^a^
Al	18.85 ± 0.78 ^a^	23.98 ± 1.49 ^b^
Ni	25.85 ± 1.21 ^a^	26.04 ± 4.26 ^a^
Zn	12.35 ± 0.39 ^a^	12.89 ± 0.44 ^a^
Na	155.33 ± 4.16 ^a^	154 ± 12.17 ^a^
Fe	309.6 ± 15.79 ^b^	218.6 ± 31.66 ^a^

Values represent mean ± SD (*n* = 3). Different letters within a row represent significant difference at *p* ≤ 0.05, derived from Student’s *t*-test.

**Table 6 jof-09-00763-t006:** Antinutrient to mineral molar ratios of *L. drassinus* and *L. controversus*.

Antinutrient to Mineral Molar Ratios	*L. drassinus*	*L. controversus*	Critical Values
OA/(Ca + Mg)	0.11	0.001	2.5 ^#^
PA/Fe	0.07	0.10	0.4 *
PA/Ca	0.02	0.03	0.2 *
PA/Zn	1.97	1.98	10 ^#^

OA—oxalic acid; PA—phytic acid. Critical values: #—Oberleas and Harland, 1981 [70]; *—Hassan et al., 2014 [71].

## Data Availability

Not available.

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
