# Peer review of "Nutritional Assessment of Lactarius drassinus and L. controversus from the Cold Desert Region of the Northwest Himalayas for Their Potential as Food Supplements"

_jof, 2023, doi:10.3390/jof9070763_

Round 1

Reviewer 1 Report

The manuscript “Nutritional profile of two wild edible mushrooms, Lactarius drassinus and L. controversus, from cold desert region of North- west Himalayas: Bioresources of nutraceutical potential” evaluates the nutritional and nutraceutical properties of two wild mushrooms to assess their potential as food supplements. This paper provides much theoretical data as a theoretical reference for these two new species of wild edible mushrooms. I think this article can be published in JoF, however, it needs major revision before that. Detailed questions are given below.

1. The references are too old, please update at least 20 references from the last five years (2019-2023), especially in the introduction and discussion sections. Recommended updated references are listed below.

DOI: 10.1016/j.biopha.2022.112621

DOI: 10.1016/j.phymed.2022.154566

DOI: 10.1016/j.ijbiomac.2021.05.139

DOI: 10.1016/j.ijbiomac.2021.01.081

DOI: 10.1080/87559129.2023.2202738

2. Line 144 has extra spaces. Please review the article thoroughly for formatting and spelling errors.

3. The data are characterized uniformly to two decimal places (Table 1).

4. Please standardize the presentation of the vertical coordinates in Figures 2 and 3 by adding "content" after the indicator, e.g. total protein content. It is suggested that the horizontal coordinate "mushroom species" could be removed and the picture could be embellished.

5. Please change the vertical coordinate in Figure 4A to the H2O2 scavenging rate (%).

None

Author Response

We are thankful to the reviewer for critically reviewing the manuscript. The suggestions and comments have been closely followed and revisions have been made accordingly. The corrections in the manuscript are highlighted in green.

The manuscript “Nutritional profile of two wild edible mushrooms, Lactarius drassinus and L. controversus, from cold desert region of North-west Himalayas: Bioresources of nutraceutical potential” evaluates the nutritional and nutraceutical properties of two wild mushrooms to assess their potential as food supplements. This paper provides much theoretical data as a theoretical reference for these two new species of wild edible mushrooms. I think this article can be published in JoF, however, it needs major revision before that. Detailed questions are given below.

  1. The references are too old, please update at least 20 references from the last five years (2019-2023), especially in the introduction and discussion sections. Recommended updated references are listed below.

DOI: 10.1016/j.biopha.2022.112621

DOI: 10.1016/j.phymed.2022.154566

DOI: 10.1016/j.ijbiomac.2021.05.139

DOI: 10.1016/j.ijbiomac.2021.01.081

DOI: 10.1080/87559129.2023.2202738

Ans. As per suggestion, we have updated the references in the introduction and discussion sections with sources from the last five years (2019-2023). By doing so, we have ensured that the most current and relevant research is reflected in these sections.

However, please note that in some cases, particularly regarding literature on Kargil, there are limitations in finding recent publications. Kargil is relatively uncharted and the availability of scholarly publications pertaining to wild mushrooms within the specified time frame specifically focusing on reports and developments are limited. Nonetheless, we have ensured that the references selected provide a comprehensive understanding of the topic.

L89-91: Contemporary researches have validated unique properties of bioactive compounds extracted from numerous species of mushrooms used in treating and reducing the severity of various medical conditions including COVID-19 [22,23].

L92-94: Mushrooms contain a wide variety of polysaccharides which are often cited for their immunomodulatory, antioxidant, antidepressant, and anticancer properties, among others [24-26].

L94-96: Clinical trials have substantiated mushrooms such as P. eryngii and A. bisporus as a favourable choice for "healthy-snacking" particularly for individuals with an unhealthy metabolism and personalized nutritional needs [27-29].

L57-59: Since the growing period is short and cultivation during winter is not possible except in greenhouses, vegetables remain a rare item in diet leading to malnourishment [3,4].

References

Guo, Y.; Chen, X.; Gong, P. Classification, Structure and Mechanism of Antiviral Polysaccharides Derived from Edible and Medicinal Fungus. Int J Biol Macromol 2021183, 1753-1773. doi: 10.1016/j.ijbiomac.2021.05.139.

Maity, P.; Sen, I.K.; Chakraborty, I.; Mondal, S.; Bar, H.; Bhanja, S.K.; Mandal, S.; Maity, G.N. Biologically Active Polysaccharide from Edible Mushrooms: A Review. Int J Biol Macromol 2021, 172, 408-417. doi: 10.1016/j.ijbiomac.2021.01.081.

Mwangi, R.W.; Macharia, J.M.; Wagara, I.N.; Bence, R.L. The Antioxidant Potential of Different Edible and Medicinal Mushrooms. Biomed Pharmacother 2022, 147, 112621. doi: 10.1016/j.biopha.2022.112621.

Zhou, Y.; Chu, M.; Ahmadi, F.; Agar, O.T.; Barrow, C.J.; Dunshea, F.R.; Suleria, H.A. A Comprehensive Review on Phytochemical Profiling in Mushrooms: Occurrence, Biological Activities, Applications and Future Prospective. Food Rev Int 2023, 1-28. doi: 10.1080/87559129.2023.2202738.

Keerthana, K.; Anukiruthika, T.; Moses, J.A.; Anandharamakrishnan, C. Development of Fiber-Enriched 3D Printed Snacks from Alternative Foods: A Study on Button Mushroom. J Food Eng 2020287, 110116. doi: 10.1016/j.jfoodeng.2020.110116.

Amerikanou, C.; Tagkouli, D.; Tsiaka, T.; Lantzouraki, D.Z.; Karavoltsos, S.; Sakellari, A.; Kleftaki, S.A.; Koutrotsios, G.; Giannou, V.; Zervakis, G.I.; Zoumpoulakis, P. Pleurotus eryngii Chips—Chemical Characterization and Nutritional Value of an Innovative Healthy Snack. Foods 2023, 12, 353. doi: 10.3390/foods12020353.

Kleftaki, S.A.; Simati, S.; Amerikanou, C.; Gioxari, A.; Tzavara, C.; Zervakis, G.I.; Kalogeropoulos, N.; Kokkinos, A.; Kaliora, A.C. Pleurotus eryngii Improves Postprandial Glycaemia, Hunger and Fullness Perception, and Enhances Ghrelin Suppression in People with Metabolically Unhealthy Obesity. Pharmacol Res 2022, 175, 105979. doi: 10.1016/j.phrs.2021.105979.

Zhou, L.W.; Ghobad-Nejhad, M.; Tian, X.M.; Wang, Y.F.; Wu, F. Current status of ‘Sanghuang’as a group of medicinal mushrooms and their perspective in industry development. Food Rev Int 2022, 38, 589-607. doi: 10.1080/87559129.2020.1740245

Tara, J.S.; Hussain, Z. Biology of Mythimna separata (Lepidoptera) on Hordeum vulgare in Arid Cold Desert of Kargil Ladakh (J&K, India). Int J Res Anal Rev 2019, 6, 320-326.

Akbar, M., Bhat, M.S. and Khan, A.A., 2023. Multi-Hazard Susceptibility Mapping for Disaster Risk Reduction in Kargil-Ladakh Region of Trans-Himalayan India. Environ Earth Sci 2023, 82, 68. doi :10.1007/s12665-022-10729-7.

Imran, M.; Ghorat, F.; Ul-Haq, I.; Ur-Rehman, H.; Aslam, F.; Heydari, M.; Shariati, M.A.; Okuskhanova, E.; Yessimbekov, Z.; Thiruvengadam, M.; Hashempur, M.H. Lycopene as a Natural Antioxidant Used to Prevent Human Health Disorders. Antioxidants 2020, 9, 706. doi: 10.3390/antiox9080706.

Nowak, R.; Nowacka-Jechalke, N.; Pietrzak, W.; Gawlik-Dziki, U. A New Look at Edible and Medicinal Mushrooms as a Source of Ergosterol and Ergosterol Peroxide-UHPLC-MS/MS Analysis. Food Chem 2022, 369, 130927. doi: 10.1016/j.foodchem.2021.130927.

Kaspchak, E.; Bonassoli, A.B.G.; Iwankiw, P.K.; Kayukawa, C.T.M.; Igarashi-Mafra, L.; Mafra, M.R. Interactions of Antinutrients Mixtures with Bovine Serum Albumin and its Influence on In Vitro Protein Digestibility. J Mol Liq 2020, 315, 113699. doi: 10.1016/j.molliq.2020.113699.

Seal, C.J.; Courtin, C.M.; Venema, K.; de Vries, J. Health Benefits of Whole Grain: Effects on Dietary Carbohydrate Quality, the Gut Microbiome, and Consequences of Processing. Compr Rev Food Sci Food Saf 2021, 20, 2742-2768. doi: 10.1111/1541-4337.12728.

Wunjuntuk, K.; Ahmad, M.; Techakriengkrai, T.; Chunhom, R.; Jaraspermsuk, E.; Chaisri, A.; Kiwwongngam, R.; Wuttimongkolkul, S.; Charoenkiatkul, S. Proximate Composition, Dietary Fibre, Beta-Glucan Content, and Inhibition of Key Enzymes Linked to Diabetes and Obesity in Cultivated and Wild Mushrooms. J Food Compos Anal 2022, 105, 104226. doi: 10.1016/j.jfca.2021.104226

Jacinto-Azevedo, B.; Valderrama, N.; Henríquez, K.; Aranda, M.; Aqueveque, P. Nutritional Value and Biological Properties of Chilean Wild and Commercial Edible Mushrooms. Food Chem 2021, 356, 129651. doi: 10.1016/j.foodchem.2021.129651.

Msomi, N.Z.; Erukainure, O.L.; Islam, M.S. Suitability of Sugar Alcohols as Antidiabetic Supplements: A Review. J Food Drug Anal 2021, 29, 1. doi: 10.38212/2224-6614.3107.

Ao, T.; Deb, C.R. Nutritional and Antioxidant Potential of Some Wild Edible Mushrooms of Nagaland, India. J Food Sci Technol 2019, 56, 1084-1089. doi: 10.1007/s13197-018-03557-w.

Nadia, J.; Bronlund, J.; Singh, R.P.; Singh, H.; Bornhorst, G.M. Structural Breakdown of Starch‐Based Foods During Gastric Digestion and its Link to Glycemic Response: In Vivo and In Vitro Considerations. Comp Rev Food Sci Food Saf 202120, 2660-2698. doi: 10.1111/1541-4337.12749.

Savarino, G.; Corsello, A.; Corsello, G. Macronutrient Balance and Micronutrient Amounts Through Growth and Development. Ital J Pediatr 2021, 47, 1-14. doi: 10.1186/s13052-021-01061-0.

González, A.; Cruz, M.; Losoya, C.; Nobre, C.; Loredo, A.; Rodríguez, R.; Contreras, J.; Belmares, R. Edible Mushrooms as a Novel Protein Source for Functional Foods. Food Funct 2020, 11, 7400-7414. doi: 10.1039/d0fo01746arsc.li/food- 636function.

Wang, M.; Zhao, R. A Review on Nutritional Advantages of Edible Mushrooms and its Industrialization Development Situation in Protein Meat Analogues. J Future Foods 2023, 3, 1-7. doi: 10.1016/j.jfutfo.2022.09.001.

Gazme, B.; Boachie, R.T.; Tsopmo, A.; Udenigwe, C.C. Occurrence, Properties and Biological Significance of Pyroglutamyl Peptides Derived from Different Food Sources. Food Sci Hum Wellness 2019, 8, 268-274. doi: 10.1016/j.fshw.2019.05.002.

Ali, Q.; Haider, M.Z.; Shahid, S.; Aslam, N.; Shehzad, F.; Naseem, J.; Ashraf, R.; Ali, A.; Hussain, S.M. Role of Amino Acids in Improving Abiotic Stress Tolerance to Plants. In Plant tolerance to environmental stress 2019, pp. 175-204. CRC Press.

Nagulwar, M.M.; More, D.R.; Mandhare, L.L. Nutritional Properties and Value Addition of Mushroom: A Review. Pharma Innov J 20209, 395-398. doi: 10.22271/tpi.2020.v9.i10f.5266.

  1. Line 144 has extra spaces. Please review the article thoroughly for formatting and spelling errors.

Ans. We agree with the reviewer’s assessment. Accordingly, the formatting and spelling errors have been reviewed thoroughly.

  1. The data are characterized uniformly to two decimal places (Table 1).

Ans. Done as suggested.

  1. Please standardize the presentation of the vertical coordinates in Figures 2 and 3 by adding "content" after the indicator, e.g., total protein content. It is suggested that the horizontal coordinate "mushroom species" could be removed and the picture could be embellished.

Ans. Done as suggested.

  1. Please change the vertical coordinate in Figure 4A to the H2O2 scavenging rate (%).

Ans. Done as suggested.

Reviewer 2 Report

- Include phylogenetic trees with your accessions for both species

- Include wild fruiting bodies production kg ha-1 for both species  

- Indicate if you obtained strains from these basidiomes

- Indicate the global distribution of both species  

Author Response

We sincerely thank the reviewer for critically reviewing the manuscript. Your insightful observations and suggestions have immensely contributed to strengthening the content of the manuscript.

Comments and Suggestions for Authors

- Include phylogenetic trees with your accessions for both species

Ans. Done as suggested.

L311-313: The phylogenetic trees were constructed using Mega11 software [69] to confirm the evolutionary relationship amongst the Lactarius spp. (Supplementary Material 2).

- Include wild fruiting bodies production kg ha-1 for both species 

Ans. The fruiting bodies production was not documented in the present study.

- Indicate if you obtained strains from these basidiomes

Ans. Strains from the basidiomes were not obtained in the present study.

- Indicate the global distribution of both species.

Ans. As per suggestion, the global distribution of Lactarius drassinus and L. controversus has been included in the revised manuscript.

L106-112: Lactarius, a genus within the Russulaceae family, exhibits an ectomycorrhizal relationship with species of Salix, Populus, and Betula [33-35]. Distribution of L. controversus spans across three continents namely, Europe, North America, and Asia. Notably, majority of the reports are from Europe (Netherlands, United Kingdom, Spain, Belgium, Serbia, Norway, Greece, Hungary, Romania), followed closely by North America (United States of America and Canada) [35-41]. In Asia, reports of L. controversus occurrence has been documented in Kazakhstan, Turkey, and India [33,43-45].

L114-116: On the other hand, L. drassinus is a recently identified species reported from Drass, Kargil by Verma et al. [46]. Hence, no biochemical investigation has been performed per se.

Reviewer 3 Report

This manuscript makes an exhaustive nutritional characterization of two wild edible mushrooms from the cold desert region of the North-west Himalayas, with promising results to be used as a daily food option. However, I still have the question of whether it is pertinent to ensure the nutraceutical potential in the title, when in this manuscript they are not addressing tests that demonstrate nutraceutical effectiveness, this potential is only hypothetically inferred by the presence of some compounds that have been declared with nutraceutical activity. Maybe they should change the title. But consider it as a suggestion.

On the other hand, I find the results obtained in the tests carried out interesting and, above all, the interpretation of the results, and that can be attractive to readers.

 In my opinion, to improve this work I make the following specific observations:

 L97-98: Mention who and where made the recent identification.

 L229: "...according to their macromorphological features." It is important that they put the source that describes the macromorphological characteristics that were used as a basis for the identification of the mushrooms.

 L344, L351, L352: "(Fig. 2)." should be (Fig. 2A), (Fig. 2B and 2C), (Fig. 2D and 2E). Please correct as appropriate for each figure call.

 L355-358: The concentration values presented in the wording "84.39 mg GAE 355 g-1 DW of total phenol, and 72.27 mg GAE g-1 DW" and "104.53 mg QE g-1 DW was present in L. controversial and 99.38 mg QE g-1 DW" do not agree with the scale of values that were presented on the 'Y' axis in figures 3A (0 to 6 mg GAE g-1 DW scale)) and 3B (0 to 0.3 mg PAEg-1 DW scale) in addition to the fact that the equivalents of the text with the figure (mg QE g-1 DW vs mg PAEg-1 DW), please verify, correct or explain why the units of measurement are completely different.

 L401: "(Fig. 3)" correct to (Fig 3C).

 L402-403: Same problem that the values described in the text do not match the scale of values and units of measurement of the 3D figure, review and correct. Also specify what it is (Fig 3D).

 L409-410: You must explain or put the citations in the table where the Critical values were obtained from.

 L419: "The present study demonstrated the health benefits" In this study, no tests were carried out to demonstrate any effect on health, only the characterization of the two fungi was carried out. Please clarify that only the characterization was carried out, because no health benefits are being scientifically demonstrated,  it is only inferred.

Author Response

We sincerely thank the team of reviewers for timely reviewing the manuscript. We have received the comments of the esteemed reviewers, and we wish to thank them for their critiques and valuable suggestions. The suggestions of reviewers have resulted in substantial improvement of the manuscript. The corrections in the manuscript are highlighted in red.

Q1. This manuscript makes an exhaustive nutritional characterization of two wild edible mushrooms from the cold desert region of the North-west Himalayas, with promising results to be used as a daily food option. However, I still have the question of whether it is pertinent to ensure the nutraceutical potential in the title, when in this manuscript they are not addressing tests that demonstrate nutraceutical effectiveness, this potential is only hypothetically inferred by the presence of some compounds that have been declared with nutraceutical activity. Maybe they should change the title. But consider it as a suggestion.

On the other hand, I find the results obtained in the tests carried out interesting and, above all, the interpretation of the results, and that can be attractive to readers.

Ans. We acknowledge that while the study did not directly demonstrate the nutraceutical effectiveness of these mushrooms, the substantial concentration of nutritional attributes, i.e., proteins, minerals, vitamins, phenolic compounds, carotenoids, 5-oxoproline, and ergosterol, among others, suggest their potential for use as a nutrient-dense food source. Considering this perspective, it is pertinent to mention the potential use of these mushrooms as food supplements in the context of nutraceuticals.

The revised title is, “Nutritional Assessment of Lactarius drassinus and L. controversus from Cold Desert Region of Northwest Himalayas for their Potential as Food Supplements”

 Q2. In my opinion, to improve this work I make the following specific observations:

 L97-98: Mention who and where made the recent identification.

Ans. Done as suggested. The revised line reads as follows

L114-116: On the other hand, L. drassinus is a recently identified species reported from Drass, Kargil by Verma et al. [46]. Hence, no biochemical investigation has been performed per se.

Q3. L292: "...according to their macromorphological features." It is important that they put the source that describes the macromorphological characteristics that were used as a basis for the identification of the mushrooms.

Ans. Done as suggested. The revised lines read as

L305-307: Species 1 was identified as Lactarius drassinus (Fig. 1A and 1B) and species 2 as Lactarius controversus (Fig. 1C and 1D) according to their macromorphological features and responses to different chemical tests confirmed according to Verma et al. [46] and Pekşen et al. [41].

Additionally, the detailed account of macromorphological identification of the mushroom species is provided in the Supplementary Material.

Q4. L344, L351, L352: "(Fig. 2)." should be (Fig. 2A), (Fig. 2B and 2C), (Fig. 2D and 2E). Please correct as appropriate for each figure call.

Ans. Done as suggested. The revised lines read as follows

L351-352: The concentrations of total protein were 293.33 and 337.78 mg g-1 DW in L. drassinus and L. controversus, respectively (Fig. 2A).

L354-357: Between the two mushrooms, L. drassinus recorded higher concentrations of TAC and TSS than L. controversus (Fig. 2B and 2C). Similarly, L. drassinus was found to contain RS and NRS in higher concentrations than L. controversus (Fig. 2D and 2E). On the contrary, starch concentrations were found higher in L. controversus than L. drassinus (Fig. 2F).

Q5.  L355-358: The concentration values presented in the wording "84.39 mg GAE 355 g-1 DW of total phenol, and 72.27 mg GAE g-1 DW" and "104.53 mg QE g-1 DW was present in L. controversial and 99.38 mg QE g-1 DW" do not agree with the scale of values that were presented on the 'Y' axis in figures 3A (0 to 6 mg GAE g-1 DW scale)) and 3B (0 to 0.3 mg PAEg-1 DW scale) in addition to the fact that the equivalents of the text with the figure (mg QE g-1 DW vs mg PAEg-1 DW), please verify, correct or explain why the units of measurement are completely different.

Ans. Due to an oversight during the submission process, the incorrect image (Fig.3) was inadvertently attached. The uploaded file was the template used for formatting and structuring the study findings. We understand that this error has led to a significant discrepancy, as the values and units of measurement in the template file are unrelated to the study itself.

We deeply regret any confusion or misunderstanding this mistake may have caused. As a responsible and committed research team, we acknowledge the error and take full responsibility for the oversight. We assure you that this incident does not reflect our usual level of diligence and attention to detail.

To rectify this issue, we have re-uploaded the correct file containing the accurate study findings. This revised submission includes all the relevant data, measurements, and results that align with the objectives of our research. We kindly request you to disregard the previously uploaded image and refer to the latest submission for an accurate understanding of our study.

 Q6. L401: "(Fig. 3)" correct to (Fig 3C).

Ans. Done as suggested. The revised lines read as follows

L384-385: Condensed tannins concentration was 0.164 mg TAE g-1 DW in L. drassinus and 0.156 mg TAE g-1 DW in L. controversus (Fig. 3C).

Q7. L402-403: Same problem that the values described in the text do not match the scale of values and units of measurement of the 3D figure, review and correct. Also specify what it is (Fig 3D).

Ans. Done as suggested. The figure has been rectified and the revised lines in the results section read as follows

L385-387: In estimation of phytic acid, L. controversus showed higher concentration (0.268 mg PAE g-1 DW), than L. drassinus (0.246 mg PAE g-1 DW) (Fig. 3D).

 Q8. L409-410: You must explain or put the citations in the table where the Critical values were obtained from.

Ans. The critical values of antinutrient and mineral molar ratios have been explained in the discussion section and read as

L467-470: Oberleas and Harland reported that foods having molar ratio [Phytate]/[Zn] less than 10 showed substantial availability of Zn [99]. Likewise, Hassan et al. reported [Phytate]/[Ca] and [Phytate]/[Fe] below 0.2 and 0.4, respectively, indicate adequate bioavailabilities of Ca and Fe [100].

L483-485: Biological interaction of Ca, Mg, and oxalate were found lower than the critical value (2.5) in the studied mushrooms [99], indicating high availabilities of Ca and Mg.

As per the suggestion, the references have been cited in the footnote of table 6 in the revised manuscript.

Critical values:

            #- Oberleas and Harland, 1981 [99]

            *- Hassan et al., 2014 [100]

 Q9. L419: "The present study demonstrated the health benefits" In this study, no tests were carried out to demonstrate any effect on health, only the characterization of the two fungi was carried out. Please clarify that only the characterization was carried out, because no health benefits are being scientifically demonstrated, it is only inferred.

Ans. As aforementioned in response to Q1, the current study endorses the potential of the collected mushroom species as locally-sourced food supplements.

L401-403: The present study characterized the nutritional and nutraceutical potential of two undocumented wild edible mushrooms, L. drassinus and L. controversus, from Kargil.

Reviewer 4 Report

The manuscript by Hom-Singli Mayirnao et al. evaluated the nutritional and nutraceutical properties of the wild edible mushrooms Lactarius drassinus and Lactarius controversus. 

Overall, the work that has been conducted is enough and although the methodologies applied are not cutting edge, the information seems novel and useful to researchers on the field.

However, this reviewer has some concerns and suggests a major revision because of the following.

1. The species investigated are wild and the evidence of potential cultivation is missing, so one would wonder why bother?

2. The authors throughout the manuscript consider their study as one that demonstrates health benefits, this should be softened.

3.The authors consider N, P, K, S, Mg, Ca as macronutrients, please correct this, these are metals, provide us with no energy and belong to micronutrients! 

4. Some very interesting citations are missing as regards the potential health effects and applications of mushrooms in nutrition for example DOI10.1016/j.phrs.2021.105979, DOI 10.3390/antiox11112113, DOI 10.3390/foods12020353 

5 Why applying Bradford instead of kjeldah?

The present study demonstrated the health benefits

A native English speaker should go through the manuscript. 

Author Response

We sincerely thank the reviewer for critically reviewing the manuscript. Your insightful observations and suggestions have immensely contributed to strengthening the content, clarity, and overall coherence of the manuscript.

The manuscript by Hom-Singli Mayirnao et al. evaluated the nutritional and nutraceutical properties of the wild edible mushrooms, Lactarius drassinus and Lactarius controversus

Overall, the work that has been conducted is enough and although the methodologies applied are not cutting edge, the information seems novel and useful to researchers on the field.

However, this reviewer has some concerns and suggests a major revision because of the following.

  • The species investigated are wild and the evidence of potential cultivation is missing, so one would wonder why bother?

Ans. The primary objective of the study was to explore and evaluate the nutritional attributes of wild edible mushrooms from Kargil. The study has provided valuable insights into their potential as a food source and exploitation of their nutritional proficiencies for the inhabitants of Kargil. Moreover, the findings of the study laid the foundation for further research and exploration in the field of mushroom cultivation. Efforts to standardize their cultivation is underway. By directing future studies towards the cultivation of these two mushrooms, their full potential can be unlocked by making them more accessible and benefiting both the local communities and the broader population.

  • The authors throughout the manuscript consider their study as one that demonstrates health benefits, this should be softened.

Ans. We have made the changes wherever necessary. The sections that were overemphasizing the health benefits have been deleted.

Mushroom carbohydrates are pharmacologically important owing to their immunomodulatory and antitumor properties [58].

Interestingly, the combination of these two amino acids is known to inhibit tumour cell proliferation of human hepatocellular carcinoma [76].

  • The authors consider N, P, K, S, Mg, Ca as macronutrients, please correct this, these are metals, provide us with no energy and belong to micronutrients! 

Ans. Corrected. The revised lines read as follows

L116-125: Considering the emerging use of wild culinary mushrooms, the present study assessed (i) metabolite profile based on Gas Chromatography-Mass Spectrometry (GC-MS); (ii) nutritional value ­ total protein, carbohydrates (total available carbohydrates, total soluble sugars, reducing sugars, non-reducing sugars, and starch), vitamins (C, B3, B6), minerals; macroelements (N, P, K, S, Mg, Ca), microelements (Na, Cu, Zn, Fe, Al, Ni), and antinutrients (phytates and tannins); and (iii) nutraceutical properties: total phenol, total flavonoid, Ultra-Performance Liquid Chromatography-based quantification of phenols (gallic acid, chlorogenic acid, vanillin, ferulic acid, and cinnamic acid), flavonoid (quercetin dihydrate), and carotenoids (lycopene and β-carotene), and reducing power as well as antioxidant potential of L. drassinus and L. controversus from Kargil.

  1. Some very interesting citations are missing as regards the potential health effects and applications of mushrooms in nutrition for example DOI10.1016/j.phrs.2021.105979, DOI 10.3390/antiox11112113, DOI 10.3390/foods12020353 

Ans. The references have been incorporated in the revised manuscript.

L89-96: Contemporary researches have validated unique properties of bioactive compounds extracted from numerous species of mushrooms used in treating and reducing the severity of various medical conditions including COVID-19 [22,23]. Mushrooms contain a wide variety of polysaccharides which are often cited for their immunomodulatory, antioxidant, antidepressant, and anticancer properties, among others [24-26]. Clinical trials have substantiated that mushrooms such as P. eryngii and A. bisporus serve as a favourable choice for "healthy-snacking" particularly for individuals with an unhealthy metabolism and personalized nutritional needs [27-29].

  1. Why applying Bradford instead of kjeldahl?

Ans. Bradford assay measures the total protein in the samples. Kjeldahl measures the protein content based on its nitrogen content whereas there may be other biomolecules containing nitrogen as well, e.g., nucleic acids and other nitrogenous compounds. Normally, the Bradford method has been recognized to be less prone to such interference.

Round 2

Reviewer 1 Report

The manuscript “Nutritional profile of two wild edible mushrooms, Lactarius drassinus and L. controversus, from cold desert region of North- west Himalayas: Bioresources of nutraceutical potential” evaluates the nutritional and nutraceutical properties of two wild mushrooms to assess their potential as food supplements. This paper provides much theoretical data as a theoretical reference for these two new species of wild edible mushrooms. The article is a significant improvement over the previous one and the authors have revised it according to the review comments. This article meets the requirements for publication.

Reviewer 4 Report

The authors have addressed all queries and the manuscript is now appropriate for publication.